# Classical Casimir free energy for two Drude spheres of arbitrary radii: A plane-wave approach

Tanja Schoger and Gert-Ludwig Ingold[⋆]

Institut für Physik, Universität Augsburg, 86135 Augsburg, Germany

⋆ gert.ingold@physik.uni-augsburg.de

## Abstract

We derive an exact analytic expression for the high-temperature limit of the Casimir interaction between two Drude spheres of arbitrary radii. Specifically, we determine the Casimir free energy by using the scattering approach in the plane-wave basis. Within a round-trip expansion, we are led to consider the combinatorics of certain partitions of the round trips. The relation between the Casimir free energy and the capacitance matrix of two spheres is discussed. Previously known results for the special cases of a sphere-plane geometry as well as two spheres of equal radii are recovered. An asymptotic expansion for small distances between the two spheres is determined and analytical expressions for the coefficients are given.

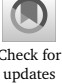
# 1  Introduction

The Casimir effect is often seen as a quantum effect arising from the vacuum fluctuations of the electromagnetic field between two objects. However, for non-zero temperature $T$ also thermal photons with wavelength $\lambda_T = \hbar c / k_B T$ contribute to the Casimir force. In fact, for distances larger than the wavelength $\lambda_T$, the main contribution to the Casimir force is due to thermal fluctuations. This leads to a finite force even in the classical limit of $\hbar \to 0$ which in view of the definition of the thermal wavelength is equivalent to the high-temperature limit $T \to \infty$. The Casimir free energy, which then no longer depends on Planck's constant, is found to be linear in temperature. Consequently, the Casimir entropy becomes constant, thereby revealing the entropic origin of the Casimir effect in the classical limit [1].

Within the scattering approach to the Casimir effect [2], the high-temperature limit amounts to taking the zero-frequency term of the Matsubara sum. The associated simplification of the problem has allowed to obtain analytical solutions not only for the archetypal plane-plane geometry [3, 4] but also for a scalar field with Dirichlet boundary conditions in the sphere-plane and sphere-sphere geometry as well as for the electromagnetic field in the sphere-plane geometry for boundary conditions corresponding to a Drude metal [5]. Even though it was suspected that the extension of the latter to two spheres of different radii might not be possible [6] we will see in the following that an analytical expression for the Casimir free energy in the general setup of two Drude spheres can be obtained within the scattering approach.

Besides the general theoretical interest in analytical solutions, there is also practical interest in such an expression. While most Casimir experiments so far have been carried out using the sphere-plane geometry, the sphere-sphere geometry has received more attention lately in experiments measuring Casimir forces [7, 8] or addressing colloidal systems [9, 10]. Carrying out the experiment in an aqueous salt solution offers the opportunity to study the zero-frequency contribution even outside the high-temperature limit by changing the salt concentration [7].

Furthermore, theoretical results in the high-temperature limit can provide a crucial ingredient to a semi-analytical approach [6] useful in the analysis of experimental data. There, the terms for non-zero Matsubara frequencies are treated within the derivative expansion. For the zero-frequency contribution, it is found to be advantageous to employ known exact results available for the sphere-plane geometry [5] and two spheres with equal radii [11]. An exact analytical expression for the setup of two spheres with arbitrary radii will thus be valuable.

It is common to treat geometries involving one or more spheres within a spherical or bispherical multipole expansion. With such approaches the high-temperature limit of two spheres with different radii has not been explicitly derived so far. However, it could have been obtained by combining the results of [5] and [12] together with the capacitance matrix discussed in Section 4.4. In [5], bispherical coordinates were employed to determine the free energy for two Dirichlet spheres while [12] applied field theoretical methods to calculate the difference in free energy of two spheres for Dirichlet and Drude boundary conditions.

Here, we will take a different approach by working in the plane-wave basis which has been shown to allow for interesting physical insights [13, 14] as well as an efficient numerical method [15]. Our derivation of the Casimir free energy of two Drude spheres with arbitrary radii will entirely be based on the plane-wave basis. A round-trip expansion of the scattering of electromagnetic waves between the two spheres leads to an interesting combinatorial problem which can be solved. Furthermore, our calculation sheds light on a relation between the scattering approach to the Casimir effect and a problem of electrostatics.

The paper is organized as follows. Section 2 introduces the scattering approach within the plane-wave basis, where we express the Casimir free energy as a sum over round trips between the two spheres. In Section 3 we illustrate the basic idea of our approach by deriving

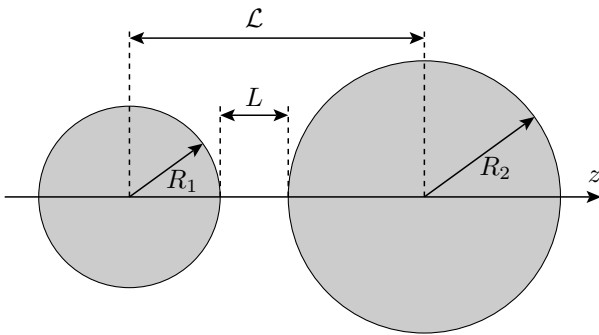

Figure 1: Representation of the geometry with two spheres of radii $R_1$ and $R_2$. The distance between the spheres is given by $L$ and $\mathcal{L}$ defines the separation between the sphere centres.

an exact expression for the Casimir free energy of a scalar field. Our result is found to be dual to the known result [5] in the sense that the free energy is obtained by summing over round trips instead of bispherical multipoles. By evaluating the spherical monopole contributions in Section 4 and subtracting them from the free energy of the scalar field, we obtain as our main result an exact expression for the Casimir free energy for an electromagnetic field in the presence of two Drude spheres. It turns out that the monopole contributions can be related to the capacitance matrix of the sphere-sphere geometry [12, 16]. Furthermore, we show that our result for the Casimir free energy agrees with previously obtained expressions for the sphere-plane geometry [5] and two spheres of equal radii [11]. Finally, in Section 5, the short-distance expansion for the general sphere-sphere geometry is derived with some technical details relegated to the appendix.

## 2 Classical Casimir free energy within the plane-wave basis

We start by compiling all ingredients required to evaluate the Casimir free energy in the high-temperature limit within the scattering approach. The geometry of our sphere-sphere setup is shown in Fig. 1 where the two spheres have generally different radii $R_1$ and $R_2$ and are placed at a centre-to-centre distance $\mathcal{L} = R_1 + R_2 + L$. $L$ denotes the smallest distance between the two sphere surfaces. The $z$-axis is chosen to go through the spheres' centres. Furthermore, the spheres are assumed to be made of a Drude-type metal with a dielectric function

$$\epsilon(i\xi) = 1 + \frac{\omega_{\mathrm{P}}^2}{\xi(\xi + \gamma)},\tag{1}$$

for imaginary frequencies $\omega = i\xi$ with the plasma frequency $\omega_{\mathrm{P}}$ and the relaxation frequency $\gamma$. The dielectric function (1) implies a finite dc conductivity $\omega_{\mathrm{P}}^2/\gamma$. As a consequence, only the electric modes contribute to the Casimir energy in the high-temperature limit as we will see below.

Within the scattering approach to the Casimir effect [2], the free energy is obtained by summation over terms containing the round-trip operator $\mathcal{M}$ at the Matsubara frequencies $\xi_n = 2\pi n/k_{\mathrm{B}}T$. Here, $k_{\mathrm{B}}$ and $T$ are the Boltzmann constant and the temperature, respectively. In the high-temperature limit $L/\lambda_T \gg 1$, only the zero-frequency term is relevant and the Casimir free energy becomes

$$\mathcal{F} = \frac{k_{\mathrm{B}}T}{2}\operatorname{tr}\log\left[1 - \mathcal{M}(\xi = 0)\right].\tag{2}$$

The round-trip operator $\mathcal{M}$ describes one complete round trip of the electromagnetic waves between the two spheres and is defined as

$$\mathcal{M} = \mathcal{R}_2 \mathcal{T}_{21} \mathcal{R}_1 \mathcal{T}_{12}. \tag{3}$$

$\mathcal{R}_1$ and $\mathcal{R}_2$ are the reflection operators for the two spheres while the operators $\mathcal{T}_{12}$ and $\mathcal{T}_{21}$ describe the translation between the centres of the spheres. In the following, we omit the argument of the round-trip operator because we will exclusively be concerned with the zero-frequency case.

For our purpose, it is convenient to expand the logarithm appearing in (2) into a Mercator series. The Casimir free energy then reads

$$\mathcal{F} = -\frac{k_B T}{2} \sum_{r=1}^{\infty} \frac{\text{tr}\mathcal{M}^r}{r}, \tag{4}$$

which in physical terms amounts to an expansion in the number $r$ of round trips.

In order to evaluate the trace in (4), we have to choose a basis. While it may appear as natural to use spherical [17–19] or bispherical [5] multipoles, we found it convenient to make use of a plane-wave basis which has been proven useful lately in the study of the sphere-sphere geometry [13, 15].

Specifically, we use the angular spectral representation [20] consisting of plane waves denoted by $|\mathbf{k}, p, \phi\rangle$. Here, $\mathbf{k}$ refers to the projection of the wave vector onto the plane perpendicular to the $z$-axis. The polarization $p$ can be transverse electric (TE) or transverse magnetic (TM) with respect to the Fresnel plane spanned by the $z$-axis and the incoming wave vector. Introducing the Wick rotated $z$-component of the wave vector $\kappa$, we obtain from the dispersion relation

$$\kappa = \left( \mathbf{k}^2 + \frac{\xi^2}{c^2} \right)^{1/2}. \tag{5}$$

Since the imaginary frequency $\xi$ is preserved during a round trip, we do not include it in the parameters characterizing the plane-wave basis. Furthermore, $\xi = 0$ in the high-temperature limit considered here, so that $\kappa = |\mathbf{k}|$. Finally, $\phi = \pm$ specifies the direction along the $z$-axis in which the plane wave decays. $\phi$ changes its sign at each reflection.

In the angular spectral representation, the trace of the $r$-th power of the round-trip operator in the plane-wave basis can now be expressed as

$$\begin{aligned} \text{tr}\mathcal{M}^r = \sum_{p_1,\dots,p_{2r}} \int \frac{d\mathbf{k}_1 \dots d\mathbf{k}_{2r}}{(2\pi)^{4r}} \prod_{j=1}^{r} e^{-\kappa_{2j}\mathcal{L}} e^{-\kappa_{2j-1}\mathcal{L}} \\ \times \langle \mathbf{k}_{2j+1}, p_{2j+1}, -|\mathcal{R}_2|\mathbf{k}_{2j}, p_{2j}, +\rangle \langle \mathbf{k}_{2j}, p_{2j}, +|\mathcal{R}_1|\mathbf{k}_{2j-1}, p_{2j-1}, -\rangle, \end{aligned} \tag{6}$$

where the indices $2r+1$ and $1$ are identified to account for the trace. The exponential factors represent the diagonal matrix elements of the two translation operators covering the distance $\mathcal{L}$ between the centres of the spheres. This latter choice allows us to make use of the standard reflection operators with the origin of the reference frame at the spheres' centres.

The expression (6) requires the knowledge of the matrix elements of the reflection operator. We concentrate on the results found in the limit of vanishing imaginary frequency $\xi$ and refer the reader to [13] for more details. The matrix elements are obtained from the Mie scattering amplitudes by transforming from the polarization basis referring to the Fresnel plane to the polarization basis referring to the scattering plane. The Mie scattering amplitudes can be expressed in terms of a sum over multipoles $\ell$ and consist of the angle functions $\tau_\ell(\cos(\Theta))$ and $\pi_\ell(\cos(\Theta))$ accounting for the scattering geometry and the material-dependent Mie coefficients $a_\ell$ and $b_\ell$ [21]. For imaginary frequencies, the scattering angle $\Theta$ is defined through $\cos(\Theta) = -c^2(\mathbf{k}_j \cdot \mathbf{k}_i + \kappa_j \kappa_i)/\xi^2$.

The low-frequency behavior of the electric Mie coefficient for spheres made of a Drude metal is given by $a_\ell \sim \xi^{2\ell+1}$. The magnetic Mie coefficient $b_\ell$ contains an additional power of $\xi$ and can thus be neglected with respect to $a_\ell$. The low-frequency behavior of the two angle functions appearing in the Mie scattering amplitudes is found as $\tau_\ell(\cos(\Theta)) \sim \xi^{-2\ell}$ and $\pi_\ell(\cos(\Theta)) \sim \xi^{-2\ell+2}$. Therefore, in the limit of vanishing $\xi$, only the combination $a_\ell \tau_\ell$ and thus only the Mie scattering amplitude for waves with polarization lying in the scattering plane contributes.

In the polarization basis taken with respect to the Fresnel plane, it follows that in the zero-frequency limit only the matrix element

$$\langle \mathbf{k}_j, \text{TM}, \pm | \mathcal{R} | \mathbf{k}_i, \text{TM}, \mp \rangle = \frac{2\pi R}{k_j} \sum_{\ell=1}^{\infty} \frac{R^{2\ell}}{(2\ell)!} \left[ 2k_i k_j \left( 1 + \cos(\varphi_i - \varphi_j) \right) \right]^\ell \tag{7}$$

differs from zero. Here, we have expressed the transverse wave vector $\mathbf{k}_i$ in polar coordinates through the modulus $k_i$ and the angle $\varphi_i$. The sum over the multipoles $\ell$ can be carried out and the non-vanishing reflection matrix becomes

$$\langle \mathbf{k}_j, \text{TM}, \pm | \mathcal{R} | \mathbf{k}_i, \text{TM}, \mp \rangle = \frac{2\pi R}{k_j} \left\{ \cosh\left[ 2R\sqrt{k_i k_j} \cos\left( \frac{\varphi_i - \varphi_j}{2} \right) \right] - 1 \right\}. \tag{8}$$

Note the subtraction of 1 because of the missing monopole term $\ell = 0$ in (7) which distinguishes the electromagnetic from the scalar case.

After inserting the reflection matrix element (8) into the expression (6) for the trace, it is convenient to switch to Cartesian coordinates $x_i = (k_i \mathcal{L})^{1/2} \cos(\varphi_i/2)$ and $y_i = (k_i \mathcal{L})^{1/2} \times \sin(\varphi_i/2)$

$$\text{tr}\mathcal{M}^r = \frac{(\rho_1 \rho_2)^r}{\pi^{2r}} \int \mathrm{d}\mathbf{x} \int \mathrm{d}\mathbf{y} \prod_{j=1}^{r} e^{-\left(x_{2j}^2 + y_{2j}^2\right)} e^{-\left(x_{2j-1}^2 + y_{2j-1}^2\right)}$$
$$\times \left[ \cosh(\chi_{2j}^{(2)}) - 1 \right] \left[ \cosh(\chi_{2j-1}^{(1)}) - 1 \right]. \tag{9}$$

Here, $\rho_n = R_n/\mathcal{L}$ denotes the dimensionless radius of spheres $n = 1, 2$ and the argument of the hyperbolic cosines is abbreviated by $\chi_i^{(n)} = 2\rho_n(x_i x_{i+1} + y_i y_{i+1})$. The trace over the $r$-th power of the round-trip operator is now given by a sum over $2r$-dimensional Gaussian integrals. After having determined the matrices associated with the bilinear forms in the exponentials, our main task will be to evaluate the corresponding determinants.

As already remarked above, the subtraction of 1 in the last two factors in (9) arises because the monopole term does not contribute in the case of electromagnetic waves. Including the monopole term amounts to considering the case of a scalar field with Dirichlet boundary conditions on the spheres. In the literature [5], it has been found useful to first evaluate the scalar case and then to determine the correction corresponding to the monopole contributions. In the next section, we will thus consider the scalar case. The plane-wave approach will lead us to an expression for the Casimir free energy which is equivalent to the known result [5].

## 3 Scalar field with Dirichlet boundary conditions

According to the discussion in the previous section, the trace over the $r$-th power of the round-trip operator for a scalar field and two spheres with Dirichlet (D) boundary conditions can be expressed in the plane-wave basis as

$$\text{tr}\mathcal{M}^r_{(\text{D})} = \frac{(\rho_1 \rho_2)^r}{(2\pi)^{2r}} \int \mathrm{d}\mathbf{x} \int \mathrm{d}\mathbf{y} \prod_{j=1}^{r} e^{-\left(x_{2j}^2 + y_{2j}^2\right)} e^{-\left(x_{2j-1}^2 + y_{2j-1}^2\right)}$$
$$\times \left[ e^{\chi_{2j}^{(2)}} + e^{-\chi_{2j}^{(2)}} \right] \left[ e^{\chi_{2j-1}^{(1)}} + e^{-\chi_{2j-1}^{(1)}} \right]. \tag{10}$$

Expanding the product, one obtains a sum over $2^{2r}$ Gaussian integrals where the bilinear form in the exponent can be written with the help of the $2r$-dimensional symmetric matrix

$$\mathbf{M}_r^{\pm} = \begin{pmatrix} 1 & \pm\rho_1 & 0 & \dots & 0 & \pm\rho_2 \\ \pm\rho_1 & 1 & \pm\rho_2 & & & 0 \\ 0 & \pm\rho_2 & 1 & \ddots & & \vdots \\ \vdots & & \ddots & \ddots & & 0 \\ 0 & & & & & \pm\rho_1 \\ \pm\rho_2 & 0 & \dots & 0 & \pm\rho_1 & 1 \end{pmatrix}, \tag{11}$$

where all combinations of signs appear in the expansion of the product in (10). As far as the determinant is concerned, it only matters whether the number of minus signs in the upper or lower half of the matrix is even or odd as indicated by the superscript $+$ or $-$ of $\mathbf{M}_r^{\pm}$, respectively.

For a single round trip, $r = 1$, the determinant reads

$$\det\mathbf{M}_1^{\pm} = 2\rho_1\rho_2\,(y \mp 1)\,, \tag{12}$$

where

$$y = \frac{1 - \rho_1^2 - \rho_2^2}{2\rho_1\rho_2} = 1 + \frac{L}{R_{\mathrm{eff}}} + \frac{L^2}{2R_{\mathrm{eff}}(R_1 + R_2)} \tag{13}$$

characterizes the geometry of the sphere-sphere arrangement with the effective radius $R_{\mathrm{eff}} = R_1 R_2/(R_1 + R_2)$. For a general number of round trips, it can be useful to view (11) as the Hamiltonian matrix of a periodic tight-binding model and to reexpress the problem in terms of transfer matrices [22, 23]. One then finds

$$\det\mathbf{M}_r^{\pm} = 2(\rho_1\rho_2)^r\,[\cosh(r\mu) \mp 1]\,, \tag{14}$$

with

$$\mu = \mathrm{arcosh}(y)\,. \tag{15}$$

We are now in a position to evaluate the Gaussian integrals in (10). Noting that the bilinear form in the exponent is given by $\mathbf{M}_r^+$ and $\mathbf{M}_r^-$ in half of the $2^{2r}$ terms each, we find with (4) the Casimir free energy for a scalar field and Dirichlet boundary conditions in the high-temperature limit as a sum over round trips

$$\mathcal{F}_{(\mathrm{D})} = -\frac{k_{\mathrm{B}}T}{2}\sum_{r=1}^{\infty}\frac{1}{2r}\frac{\cosh(r\mu)}{\sinh^2(r\mu)}\,. \tag{16}$$

Our result (16) can be viewed as a dual representation of the earlier result presented in [5]. Following the notation introduced there, we define

$$Z = \exp(-\mu) \tag{17}$$

and write the Casimir free energy as

$$\mathcal{F}_{(\mathrm{D})} = -\frac{k_{\mathrm{B}}T}{2}Z\frac{\mathrm{d}}{\mathrm{d}Z}\sum_{r=1}^{\infty}\frac{1}{r^2}\frac{Z^r}{1 - Z^{2r}}\,. \tag{18}$$

Expanding the last factor in a geometric series, the sum over $r$ can be evaluated. With the help of the Mercator series, one finally obtains

$$\mathcal{F}_{(\mathrm{D})} = \frac{k_{\mathrm{B}}T}{2}\sum_{l=0}^{\infty}(2l + 1)\log(1 - Z^{2l+1})\,, \tag{19}$$

in agreement with the result derived by means of bispherical coordinates [5].

Particularly for small distances, the round-trip representation (16) may be numerically advantageous as compared to the expansion (19) because it possesses better convergence properties. It is also straightforward to read off the Casimir free energy within the proximity force approximation by simply retaining the leading order of the hyperbolic functions. Details of the asymptotic expansion in $\mu$ of the Casimir free energy will be discussed in section 5.

# 4 Electromagnetic case for two general Drude spheres

## 4.1 Monopole contributions in the scalar case

The trace over the $r$-th power of the round-trip operator in the electromagnetic case differs from the scalar case only by the monopole term $\ell = 0$ as one can see by comparing the corresponding expressions (9) and (10). As already discussed at the end of section 2, does the $-1$ in the brackets of (9) account for the subtraction of the monopole term. Hence, when expanding the product in (9), all summands containing at least one of these terms $-1$ yield the negative monopole contributions present in the scalar case. In the following, we will thus focus on the difference

$$\Delta_r = \operatorname{tr} \mathcal{M}^r - \operatorname{tr} \mathcal{M}^r_{(\mathrm{D})}. \tag{20}$$

Already in previous works it was found convenient to study the difference between the scalar case with Dirichlet boundary conditions and the electromagnetic case for Drude-type objects [5, 12].

The difference $\Delta_r$ consists of a sum over Gaussian-type integrals where the bilinear form in the exponent is represented by a tridiagonal matrix with the off-diagonal matrix elements arising from the hyperbolic cosines. Whenever in the expansion of the product in (9) a factor $-1$ appears, the corresponding pair of off-diagonal matrix elements vanishes. In contrast to the matrix (11) in the scalar case, the matrix representing the bilinear form in the exponent of the integrand in (9) is now block-diagonal and can be written as

$$\mathbf{M}_w = \operatorname{diag}\left(\mathbf{m}_{n_1}^{(t_1)} \mathbf{m}_{n_2}^{(t_2)} \mathbf{m}_{n_3}^{(t_3)} \dots \mathbf{m}_{n_k}^{(t_k)}\right), \tag{21}$$

where $w$ denotes an element of a set $\Pi_{2r,k}$ containing a multiset of tuples $\{(n_1, t_1), (n_2, t_2), (n_3, t_3), \dots, (n_k, t_k)\}$ with $\sum_i n_i = 2r$ for $r$ round trips. Each block is a symmetric tridiagonal 2-Toeplitz matrix [24] of the form

$$\mathbf{m}_n^{(1/2)} = \begin{pmatrix} 1 & \pm\rho_{1/2} & \cdots & & 0 \\ \pm\rho_{1/2} & 1 & \pm\rho_{2/1} & & \\ \vdots & \pm\rho_{2/1} & 1 & \ddots & \\ & & \ddots & \ddots & \\ 0 & & & & 1 \end{pmatrix}, \tag{22}$$

where pairs of off-diagonal matrix elements alternate between $\rho_1$ and $\rho_2$ and each pair can come with an arbitrary sign. Each block is characterized by its size $n$ and the index of the first off-diagonal entry, indicated by the superscript 1 or 2 and accounted for by $t_i$ in the multiset $w$. As we will see in more detail later, we cannot set $t_i$ to 1 or 2 freely. Rather, its value needs to be compatible with the values of $n_{i-1}$ and $t_{i-1}$.

The result of the Gaussian integration will involve the determinant of $\mathbf{M}_w$ which equals the product of the determinants of the individual blocks. For odd dimension, the determinant of the blocks (22) is given by [25]

$$\det \mathbf{m}_{2k+1}^{(1/2)} = (\rho_1 \rho_2)^k U_k(y), \tag{23}$$

while for even dimension one has to distinguish between blocks starting with $\rho_1$ or $\rho_2$ on the off-diagonal

$$\det \mathbf{m}^{(1/2)}_{2k} = (\rho_1 \rho_2)^k \left[ U_k(y) + \frac{\rho_{2/1}}{\rho_{1/2}} U_{k-1}(y) \right]. \tag{24}$$

Here, $U_k$ denotes the Chebyshev polynomial of the second kind and order $k$ [26] while $y$ has been introduced in (13) and characterizes the geometry of the sphere-sphere setup. We note that the determinants do not depend on the choice of signs in (22) so that in view of the Gaussian integration we end up with $2^{n-1}$ equivalent blocks $\mathbf{m}^{(1/2)}_n$.

For the monopole contributions (20) we now obtain together with (9) and (10)

$$\Delta_r = \frac{(\rho_1 \rho_2)^r}{\pi^{2r}} \sum_{k=1}^{2r} (-1)^k \sum_{w \in \Pi_{2r,k}} \int \mathrm{d}\mathbf{x}\, e^{-\mathbf{x}^t \mathbf{M}_w \mathbf{x}} \int \mathrm{d}\mathbf{y}\, e^{-\mathbf{y}^t \mathbf{M}_w \mathbf{y}}, \tag{25}$$

where the number of blocks in $\mathbf{M}_w$ is given by the summation index $k$ and $\mathbf{x}^t$ denotes the transpose of $\mathbf{x}$. In the derivation, we have taken into account that a factor $-2$ is associated with each block as one can see by evaluating the product in (9). Furthermore, we have accounted for the multiplicity related to the signs in the block matrices mentioned above. Evaluating the Gaussian integrals, we arrive at

$$\Delta_r = (\rho_1 \rho_2)^r \sum_{k=1}^{2r} (-1)^k \sum_{w \in \Pi_{2r,k}} \frac{1}{\det \mathbf{M}_w}. \tag{26}$$

## 4.2 Combinatorics of blocks and diagrammatic representation

The decomposition of the block matrix $\mathbf{M}_w$ into blocks can be conveniently analyzed in terms of a diagrammatic representation. For $r$ round trips, we consider a graph consisting of a chain of $2r+1$ nodes where the last node should be identified with the first one. These nodes represent the two spheres and are depicted successively in black and white corresponding to spheres 1 and 2, respectively. Each block $\mathbf{m}^{(1/2)}_n$ is represented by a black or white line where the colors refer to the superscripts 1 and 2, respectively. Therefore, the color of a line equals the color of the node from which the line starts, reading the diagram from left to right. The length of the line is given by the dimension $n$ of the corresponding block.

The connection between the block matrix and its diagrammatic representation is illustrated in Fig. 2. In this example, the round trip starts on sphere 1 and the block $\mathbf{m}^{(1)}_1$ represents half a round trip ending on sphere 2. Given the odd dimension of the first block, the color switches to white. The next block, $\mathbf{m}^{(2)}_3$, has again an odd dimension and corresponds to one and a half round trips. In general, it is not required that matrices of odd dimension follow each other directly. We are now back to sphere 1 and it follows a black line symbolizing the block $\mathbf{m}^{(1)}_2$, i.e. a single round trip. This example illustrates why the values of a tuple $(n_i, t_i)$ in the multiset $w$ depend on the values of the preceding tuple as mentioned earlier.

Disregarding the color for a second, there exists an obvious connection to the problem of integer partition. It is well known that so-called ordinary Bell polynomials defined through

$$\left( \sum_{i=1}^{\infty} c_i x^i \right)^k = \sum_{n=k}^{\infty} \hat{B}_{n,k}(c_1, c_2, \ldots) x^n \tag{27}$$

provide such a partition [27]. We will have the opportunity to employ this relation later when deriving the short-distance behavior of the Casimir free energy. However, for the following discussion we will need to keep the color because according to (24) the determinant of our blocks for even dimension depends on the superscript.

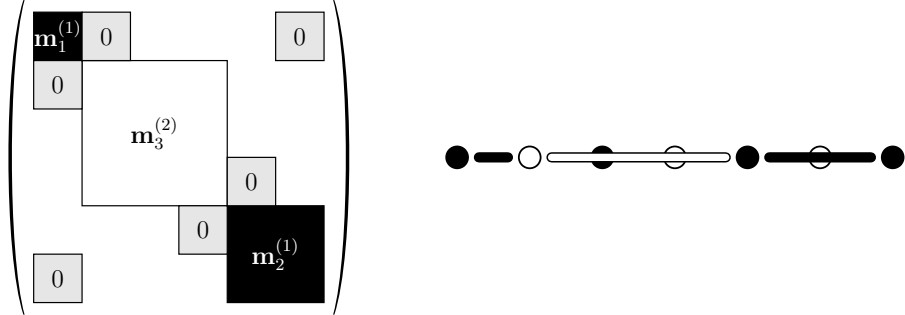

Figure 2: An example for a block matrix associated with a term contained in the monopole contributions (26) for $r = 2$, $k = 3$ is shown on the left together with the corresponding diagrammatic representation depicted on the right-hand side of the figure. In the diagram, the color of a line is determined by the color of the node to its left.

To the best of our knowledge, a generalization of the Bell polynomials to our situation with color does not exist. Fortunately, the dependence of the color of a line on the length and color of the previous line allows us to express the partitions in a recursive way instead.

Let us consider $r$ round trips, i.e. a chain of length $2r$. We introduce functions $h_{2r}^{(1)}$ and $h_{2r}^{(2)}$ as sums over the inverse determinants of all possible block matrices for $r$ round trips starting on spheres 1 and 2, respectively. These functions will allow us later to express the monopole contributions $\Delta_r$ as given by (26). For convenience, we define abbreviations for the inverse of the determinants of the blocks with $n \geq 1$

$$a_n = \frac{1}{\det \mathbf{m}_n^{(1)}}, \quad b_n = \frac{1}{\det \mathbf{m}_n^{(2)}}, \tag{28}$$

which will occur in $\Delta_r$. In our diagrams, the coefficients $a_n$ are thus associated with black lines and the coefficients $b_n$ with white ones.

We can now express the first of the two required functions, which starts on sphere 1, recursively as

$$h_{2r}^{(1)}(t) = t a_{2r} - \sum_{n=1}^{r-1} t a_{2n} h_{2(r-n)}^{(1)}(t) - \sum_{n=1}^{r} t a_{2n-1} h_{2(r-n)+1}^{(2)}(t), \tag{29}$$

where the variable $t$ will later serve to determine the number of blocks. The first term on the right-hand side of (29) accounts for a single block of maximal size while the other two terms correspond to matrices consisting of more than one block. The second term arises from a single block of even dimension followed by a block matrix starting again from sphere 1. The sum runs over all possible sizes of the first block. The relative minus sign between the first and the second term is due to the fact that each new block contributes a minus sign leading to the factor $(-1)^k$ for $k$ blocks in (26). The third term differs from the second one in so far as the first block has an odd dimension so that the remaining part starts on sphere 2. Instead of $h^{(1)}$ in the second term, we thus have $h^{(2)}$ in the third term.

To close the system of recursive equations, we similarly derive three more equations

$$h_{2r+1}^{(1)}(t) = t a_{2r+1} - \sum_{n=1}^{r} t a_{2n} h_{2(r-n)+1}^{(1)}(t) - \sum_{n=0}^{r-1} t a_{2n+1} h_{2(r-n)}^{(2)}(t) \tag{30}$$

and for graphs starting on sphere 2

$$h_{2r}^{(2)}(t) = t b_{2r} - \sum_{n=1}^{r-1} t b_{2n} h_{2(r-n)}^{(2)}(t) - \sum_{n=1}^{r} t b_{2n-1} h_{2(r-n)+1}^{(1)}(t) \tag{31}$$

and

$$h_{2r+1}^{(2)}(t) = t\,b_{2r+1} - \sum_{n=1}^{r} t\,b_{2n} h_{2(r-n)+1}^{(2)}(t) - \sum_{n=0}^{r-1} t\,b_{2n+1} h_{2(r-n)}^{(1)}(t). \tag{32}$$

We note in passing that these recursion relations can be interpreted as the Laplace expansion of appropriately chosen Hessenberg matrices but we will not make use of this fact in the following.

The sum $h_{2r}^{(1)} + h_{2r}^{(2)}$ accounts for all different kinds of block matrices occurring for $r$ round trips. However, we have not yet properly accounted for the multiplicity of the block matrices. So far, we have considered open-chain diagrams with the starting point chosen at a specific node for which a pair of off-diagonal elements vanishes. Since the trace at the origin of (26) implies a closed chain, we can have several starting points. As Fig. 3 demonstrates for $r = 3$, a configuration can start at three different nodes. Specifically, the white line can start on one of the three white nodes. Since, in general, there are $r$ nodes of one color, we have to multiply the contribution for $r$ round trips by a factor of $r$.

By proceeding as just described, we include all circular permutations of the blocks, but the functions $h_{2r}^{(1,2)}$ already include all non-equal circular permutations. Hence, to avoid double counting, we have to remove the $k$ cyclic permutations of a partition in $k$ blocks by dividing the contribution arising from $k$ blocks by $k$. Fig. 3 illustrates a non-trivial example. On the left, we present the diagrams of the partition $\mathrm{diag}(\mathbf{m}_2^{(1)}\mathbf{m}_1^{(1)}\mathbf{m}_2^{(2)}\mathbf{m}_1^{(2)})$ and all its circular permutations. As demonstrated by the graphs on the right, there are $r = 3$ possible ways of choosing a starting point for each partition. However, as one can see, each diagram appears four times on the right. Hence, by dividing by four, the correct number of block matrices is obtained.

We now make use of the parameter $t$ introduced in equations (29)–(32) which ensures that contributions arising from $k$ blocks come with a factor $t^k$. The monopole contributions for a given number of round trips (26) can thus be written as

$$\Delta_r = -(\rho_1 \rho_2)^r r \int_0^1 \mathrm{d}t\, \frac{h_{2r}^{(1)}(t) + h_{2r}^{(2)}(t)}{t}. \tag{33}$$

The negative sign arises due to our definition of $h_{2r}^{(1,2)}$, where all partitions in odd numbers of blocks occur with a positive sign compared to those with even numbers.

So far, we have considered the monopole contributions for a given number of round trips. The full correction of the Casimir free energy with respect to the result (16) or (19) for the scalar case is obtained by summation over all numbers $r$ of round trips, which we will carry out in the following section.

### 4.3 Monopole correction to the classical Casimir free energy

According to the round-trip expansion (4) of the Casimir free energy, the monopole contributions are determined by

$$\Delta = \mathcal{F} - \mathcal{F}_{(D)} = -\frac{k_B T}{2} \sum_{r=1}^{\infty} \frac{\Delta_r}{r}. \tag{34}$$

Strictly speaking, $\Delta$ equals the negative monopole contributions, but for simplicity we will continue to refer to this quantity as monopole contributions.

After inserting (33) and interchanging summation and integration, it is convenient to introduce the generating functions for the inverse block-matrix determinants

$$H^{(1,2)}(x;t) = \sum_{n=1}^{\infty} h_n^{(1,2)}(t) x^n = H_e^{(1,2)}(x;t) + H_o^{(1,2)}(x;t). \tag{35}$$

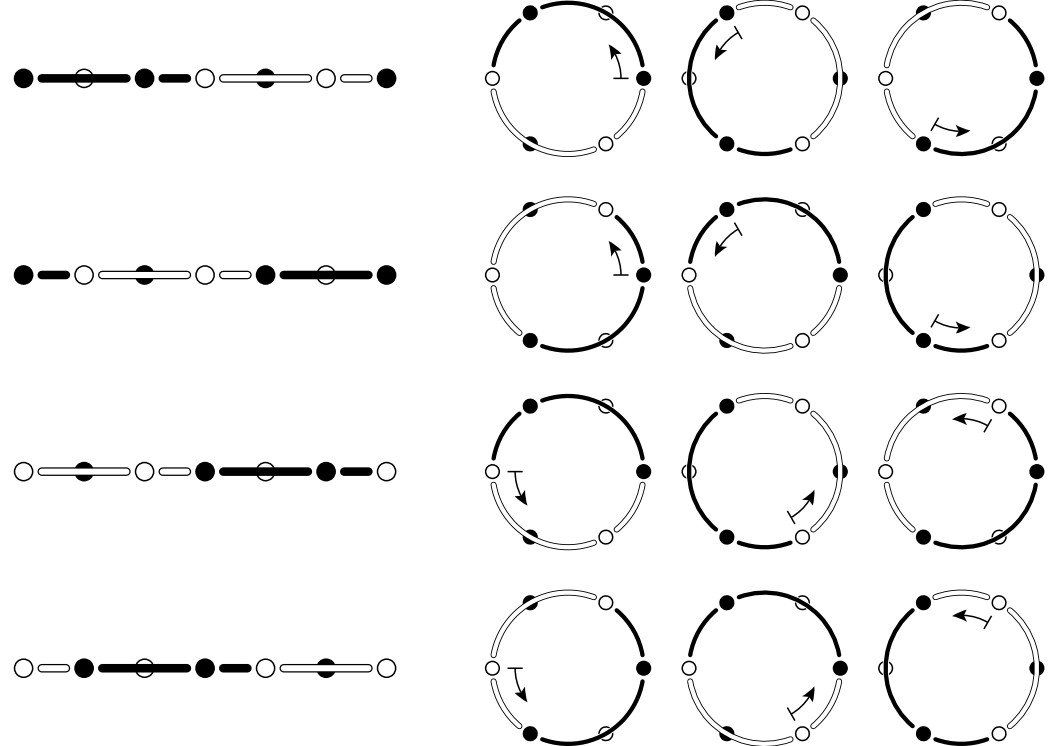

Figure 3: Diagrams for the configurations corresponding to $\mathrm{diag}(\mathbf{m}_2^{(1)}\mathbf{m}_1^{(1)}\mathbf{m}_2^{(2)}\mathbf{m}_1^{(2)})$, $\mathrm{diag}(\mathbf{m}_1^{(1)}\mathbf{m}_2^{(2)}\mathbf{m}_1^{(2)}\mathbf{m}_2^{(1)})$, $\mathrm{diag}(\mathbf{m}_2^{(2)}\mathbf{m}_1^{(2)}\mathbf{m}_2^{(1)}\mathbf{m}_1^{(1)})$ and $\mathrm{diag}(\mathbf{m}_1^{(2)}\mathbf{m}_2^{(1)}\mathbf{m}_1^{(1)}\mathbf{m}_2^{(2)})$ from top to bottom. The diagrams on the right account for the three possible starting points of each partition, when considering a closed chain. The arrows indicate the starting points of the partitions. By circularly permuting $\mathrm{diag}(\mathbf{m}_2^{(1)}\mathbf{m}_1^{(1)}\mathbf{m}_2^{(2)}\mathbf{m}_1^{(2)})$, we reproduce all four configurations. On a closed chain the four permutations appear naturally so that each diagram occurs four times on the right-hand side.

In the second equality, we decompose the sum into contributions from even (e) and odd (o) powers of $x$. Hence, the monopole term yields

$$\Delta = \frac{k_{\mathrm{B}}T}{2}\int_0^1 \mathrm{d}t\, \frac{H_{\mathrm{e}}^{(1)}(\sqrt{\rho_1\rho_2};t) + H_{\mathrm{e}}^{(2)}(\sqrt{\rho_1\rho_2};t)}{t}. \tag{36}$$

Correspondingly, we introduce the generating functions for the inverse determinants of individual blocks (28)

$$A(x) = \sum_{n=1}^{\infty} a_n x^n = A_{\mathrm{e}}(x) + A_{\mathrm{o}}(x) \tag{37}$$

and

$$B(x) = \sum_{n=1}^{\infty} b_n x^n = B_{\mathrm{e}}(x) + B_{\mathrm{o}}(x), \tag{38}$$

where we apply the same decomposition of the functions as in (35).

The generating functions $H_{\mathrm{e}}^{(1,2)}$ can be determined by summing over the recurrence relations (29)–(32) and we find

$$H_{\mathrm{e}}^{(1)}(x;t) = t\frac{A_{\mathrm{e}}(x) + tA_{\mathrm{e}}(x)B_{\mathrm{e}}(x) - tA_{\mathrm{o}}(x)B_{\mathrm{o}}(x)}{(1 + tA_{\mathrm{e}}(x))(1 + tB_{\mathrm{e}}(x)) - t^2 A_{\mathrm{o}}(x)B_{\mathrm{o}}(x)}, \tag{39}$$

with an analogous expression for $H_{\mathrm{e}}^{(2)}$ where the functions $A$ and $B$ are interchanged. The sum of both even functions yields

$$H_{\mathrm{e}}^{(1)}(x;t) + H_{\mathrm{e}}^{(2)}(x;t) = t\frac{A_{\mathrm{e}}(x) + B_{\mathrm{e}}(x) + 2tA_{\mathrm{e}}(x)B_{\mathrm{e}}(x) - 2tA_{\mathrm{o}}(x)B_{\mathrm{o}}(x)}{(1+tA_{\mathrm{e}}(x))(1+tB_{\mathrm{e}}(x)) - t^2A_{\mathrm{o}}(x)B_{\mathrm{o}}(x)}. \tag{40}$$

Noting that the numerator equals the derivative with respect to $t$ of the denominator, it is straightforward to evaluate the integral in (36) and we find

$$\Delta = \frac{k_{\mathrm{B}}T}{2}\log\left[\left(1 + A_{\mathrm{e}}(\sqrt{\rho_1\rho_2})\right)\left(1 + B_{\mathrm{e}}(\sqrt{\rho_1\rho_2})\right) - A_{\mathrm{o}}(\sqrt{\rho_1\rho_2})B_{\mathrm{o}}(\sqrt{\rho_1\rho_2})\right] \tag{41}$$

with

$$A_{\mathrm{e}}(\sqrt{\rho_1\rho_2}) = \sum_{n=1}^{\infty}\frac{1}{U_n(y) + \alpha U_{n-1}(y)}, \tag{42}$$

$$B_{\mathrm{e}}(\sqrt{\rho_1\rho_2}) = \sum_{n=1}^{\infty}\frac{1}{U_n(y) + \beta U_{n-1}(y)}, \tag{43}$$

$$A_{\mathrm{o}}(\sqrt{\rho_1\rho_2}) = \sqrt{\rho_1\rho_2}\sum_{n=0}^{\infty}\frac{1}{U_n(y)} = B_{\mathrm{o}}(\sqrt{\rho_1\rho_2}), \tag{44}$$

where $\alpha = R_2/R_1$ and $\beta = R_1/R_2$ take the ratios of the sphere radii into account and $y$ is defined in (13). The expressions in (42)–(44) are obtained by inserting (28) together with (23) and (24) into (37) and (38).

It is instructive to convince oneself that indeed all partitions of round trips are contained in (41) by expanding the logarithm as

$$\Delta = -\frac{k_{\mathrm{B}}T}{2}\left\{\sum_{k=1}^{\infty}\frac{(-1)^k}{k}\left(A_{\mathrm{e}}^k + B_{\mathrm{e}}^k\right) + \sum_{n=1}^{\infty}\frac{1}{n}\left[\sum_{l=0}^{\infty}(-A_{\mathrm{e}})^l A_{\mathrm{o}}\sum_{m=0}^{\infty}(-B_{\mathrm{e}})^m B_{\mathrm{o}}\right]^n\right\}. \tag{45}$$

The first sum accounts for arbitrary repetitions of full round trips starting either on sphere 1 or on sphere 2 as represented by $A_{\mathrm{e}}$ or $B_{\mathrm{e}}$, respectively. Expressions containing both $A_{\mathrm{e}}$ and $B_{\mathrm{e}}$ can only arise if half round trips represented by $A_{\mathrm{o}}$ and $B_{\mathrm{o}}$ occur as is the case in the second term. Reading this term from left to right, it can clearly be seen that half a round trip induces a change between full round trips starting on sphere 1 and on sphere 2. The number of factors $(-1)$ correctly reflects the number of blocks in the matrices $\mathbf{M}_w$.

## 4.4 Relation to the capacitance matrix

It appears that the result (41) was so far not known in the Casimir community. Nevertheless, it can be obtained by combining results from the literature, a fact which we only became aware of after the work presented here had been carried out. As was shown by Fosco *et al.* [12], the difference between the Casimir free energy of objects made of Drude metals and the Casimir free energy for a scalar field with Dirichlet boundary conditions is related to the capacitance matrix $\mathbf{C}$ of the arrangement of conductors. For the special case of two conductors, [12] found[1]

$$\Delta = \frac{T}{2}\log\left[\det(\mathbf{C})T^2\right]. \tag{46}$$

Even though this was not mentioned in [12], the capacitance matrix elements of two conducting spheres of arbitrary radii were already known to Maxwell [28]. Following the more

---

[1]Note that here we adopt the choice of units of [12]. Furthermore, their quantity $\Delta F$ equals $-\Delta$.

modern notation in [16], the capacitance coefficients can be expressed as

$$c_{11} = R_1(1 + B_e(\sqrt{\rho_1 \rho_2})), \tag{47}$$

$$c_{22} = R_2(1 + A_e(\sqrt{\rho_1 \rho_2})), \tag{48}$$

$$c_{12} = c_{21} = -\sqrt{R_1 R_2} A_o(\sqrt{\rho_1 \rho_2}). \tag{49}$$

Comparing these coefficients and (46) with our result (41) connects the capacitance coefficients to the scattering of electromagnetic waves in the static limit. It thus highlights the relation between our round-trip description and the method of image charges used by [28] to obtain the capacitance coefficients.

We remark that the general result (46) and our result (41) differ by a factor $R_1 R_2 T^2$ in the logarithm. While this factor would be irrelevant for the Casimir force, it makes a difference for the Casimir entropy. Its origin can be traced back to the different handling of the Casimir free energy of the individual objects [1, 29]. While the scattering approach does not contain the free energy of the spheres at an infinite distance, this contribution is present in [12]. For (41), the entropy in the high-temperature limit becomes a constant as expected [1].

### 4.5 Casimir free energy for two Drude spheres of general radii and limiting cases

According to (34), the sum of the expressions (16) and (41) gives the Casimir free energy for two Drude spheres of arbitrary radii and thus constitutes the main result of this paper. Instead of reproducing the two expressions here, it is useful to resum the result as we did in Section 3 for the scalar case and to express it in terms of the variable $Z$ introduced in (17). Noting that the Chebyshev polynomials of the second kind appearing in equations (42)–(44) can be written as [26]

$$U_n(y) = \frac{Z^{-(n+1)} - Z^{n+1}}{Z^{-1} - Z}, \tag{50}$$

we obtain the classical Casimir free energy for two Drude spheres as

$$
\begin{aligned}
\mathcal{F} = \frac{k_B T}{2} \Bigg\{ & \sum_{l=0}^{\infty} (2l+1) \log(1 - Z^{2l+1}) \\
& + \log \Bigg[ \left( 1 + \frac{1 - g_\alpha(Z)^2}{g_\alpha(Z)} \sum_{l=0}^{\infty} \frac{(Z g_\alpha(Z))^{2l+1}}{1 - Z^{2l+1}} \right) \\
& \quad \times \left( 1 + \frac{1 - g_\beta(Z)^2}{g_\beta(Z)} \sum_{l=0}^{\infty} \frac{(Z g_\beta(Z))^{2l+1}}{1 - Z^{2l+1}} \right) \\
& \quad - \frac{(1 - g_\alpha(Z)^2)(1 - g_\beta(Z)^2)}{Z} \left( \sum_{l=0}^{\infty} \frac{Z^{2l+1}}{1 - Z^{2l+1}} \right)^2 \Bigg] \Bigg\},
\end{aligned} \tag{51}
$$

where we have introduced the function

$$g_\alpha(Z) = \left( \frac{Z^2 + \alpha Z}{1 + \alpha Z} \right)^{1/2} \tag{52}$$

and correspondingly for $g_\beta(Z)$, where $\alpha$ and $\beta = 1/\alpha$ are the ratios of sphere radii as defined below (44).

We obtain the limit of a sphere of radius $R$ in front of a plane by setting $R_2 = R$ and letting $R_1$ go to infinity. Then, $g_\beta = 1$ and $B_e$ and $B_o$ vanish because there is no second sphere where

the electromagnetic waves could be scattered. Since the functional dependence of the scalar part of (51), i.e. the first sum, is not affected, we focus on the monopole contributions for which we obtain

$$
\begin{aligned}
\Delta^{(R_1 \to \infty)} &= \frac{k_B T}{2} \log(1 + A_e) \\
&= \frac{k_B T}{2} \log\left[ 1 + (1 - Z^2) \sum_{l=0}^{\infty} \frac{Z^{4l+1}}{1 - Z^{2l+1}} \right],
\end{aligned}
\tag{53}
$$

where $Z$ depends only on the aspect ratio $\epsilon = L/R$ through

$$
Z = 1 + \epsilon - \sqrt{\epsilon(2 + \epsilon)}.
\tag{54}
$$

By some minor transformations, one can convince oneself, that (53) agrees with the result found earlier by Bimonte and Emig [5].

Similarly, we obtain the Casimir free energy for two Drude spheres of equal radii by setting $R_1 = R_2 = R$ so that $g_\alpha = g_\beta = Z^{1/2} = Y$. In this case, the scattering at the two spheres cannot be distinguished and we have $A_e = B_e$ and $A_o = B_o$. The monopole contributions then read

$$
\begin{aligned}
\Delta^{(R_1 = R_2)} &= \frac{k_B T}{2} \left[ \log(1 + A_e + A_o) + \log(1 + A_e - A_o) \right] \\
&= \frac{k_B T}{2} \left[ \log\left( 1 - \sum_{l=1}^{\infty} \frac{(1 - Y^2)(1 - Y^{2l}) Y^{2l+1}}{1 - Y^{2l+1}} \right) \right. \\
&\quad + \log\left( 1 + \sum_{l=1}^{\infty} \frac{(1 - Y^2)(1 - Y^{2l}) Y^{2l+1}}{1 + Y^{2l+1}} \right) \\
&\quad \left. - \log(1 - Y^2) \right],
\end{aligned}
\tag{55}
$$

where the parameter $Y$ is a function of the aspect ratio $\delta = L/2R$

$$
Y = 1 + \delta - \sqrt{\delta(2 + \delta)}.
\tag{56}
$$

The result (55) leads to the same Casimir free energy as obtained earlier by using the transformation optics approach [11].

## 5 Short-distance expansion

In experiments, the closest distance $L$ between the two spheres is typically small compared to the radii $R_1$ and $R_2$. Therefore, we will now determine a short-distance expansion of the Casimir free energy (51) by separately considering the scalar contribution $\mathcal{F}_{(D)}$ and the monopole contributions $\Delta$. The leading-order term will correspond to the proximity-force approximation whose validity can be assessed by the higher-order terms.

In the following, we make use of the fact that $\mathcal{F}_{(D)}$ and $\Delta$ can be expressed in terms of the F-series introduced by Garvin [30] as a generalization of the Lambert series. With a choice of coefficients appropriate for our situation, we introduce

$$
\mathcal{L}_q(s, x) = \sum_{k=1}^{\infty} \frac{k^s q^{kx}}{1 - q^k}.
\tag{57}
$$

Here, we follow the notation used by Banerjee and Wilkerson who provide an asymptotic expansion of this series around $q = 1$ [31].

We start by expanding the Casimir free energy for two Dirichlet spheres. Its representation (18) can be expressed in terms of the series (57) as

$$\mathcal{F}_{(\text{D})} = -\frac{k_{\text{B}}T}{2} Z \frac{\text{d}}{\text{d}Z} \left[ \mathcal{L}_Z(-2,1) - \mathcal{L}_{Z^2}(-2,1) \right]. \tag{58}$$

For small distances $L \ll R_1, R_2$, the variable $Z = \exp(-\mu)$ is close to unity and $\mu$ defined in (15) is small. In the following, we will use $\mu$ as our expansion variable.

Making use of the asymptotic expansion of the generalized Lambert series around $q = 1$ stated in theorem 2.2 of [31], we obtain from (58) for the Casimir free energy in the scalar case with Dirichlet boundary conditions

$$\mathcal{F}_{(\text{D})} = \frac{k_{\text{B}}T}{2} \left[ -\frac{\zeta(3)}{2\mu^2} + \frac{1}{12}\log(\mu) + \frac{1}{12} - \log(A) + \frac{1}{6}\log(2) \right. \\ \left. + \sum_{n=1}^{\infty} \frac{2n+1}{2n} \frac{B_{2n}B_{2n+2}}{(2n+2)!} \left( 2^{2n+1} - 1 \right) \mu^{2n} \right], \tag{59}$$

with Glaisher's constant $A = 1.28242\ldots$ and the Bernoulli numbers $B_k$ [26]. The first term corresponds to the high-temperature result of the proximity-force approximation with the specific value of the Riemann zeta function $\zeta(3) = 1.20205\ldots$ The terms up to order $\mu^4$ were already given in [5] and are consistent with our result.

Now we turn to the short-distance expansion of the monopole term as given by the second term of (51) where the argument of the logarithm can again be expressed in terms of a generalized Lambert series (57). Readers not interested in the technical details of the derivation will find the final result in (77).

We bring the sums in the monopole contributions $\Delta$ into the form of a generalized Lambert series by writing the function (52) as

$$g_{\alpha} = Z^{1/2 + v(\mu)}, \tag{60}$$

where

$$v(\mu) = \frac{1}{2} - \frac{1}{2\mu} \left[ \log(1 + \alpha e^{\mu}) - \log(1 + \alpha e^{-\mu}) \right]. \tag{61}$$

Replacing $\alpha$ by $\beta$ simply changes the sign of this function, so that $g_{\beta} = Z^{1/2-v}$. For our purpose, we need to expand $v(\mu)$ into a Taylor series

$$v(\mu) = \sum_{n=0}^{\infty} v_n \mu^{2n}. \tag{62}$$

The coefficients for $n > 0$ can be expressed in terms of

$$v_0 = \frac{1}{2} \frac{R_1 - R_2}{R_1 + R_2} \tag{63}$$

as

$$v_n = \frac{1}{(2n+1)!} \sum_{k=0}^{2n} k! S(2n+1, k+1) (v_0 - 1/2)^{k+1}, \tag{64}$$

where $S(n, k)$ are the Stirling numbers of the second kind.

Before making use of the generalized Lambert series, it is convenient to introduce a notation for the prefactors and sums appearing in the argument of the second logarithmic term in (51). By defining

$$J(c) = \frac{1 - Z^{2(c-1)}}{Z^{c-1}} \tag{65}$$

and

$$I(c) = \sum_{l=0}^{\infty} \frac{Z^{c(2l+1)}}{1 - Z^{2l+1}}, \tag{66}$$

the monopole contributions (41) can be brought into the form

$$\Delta = \frac{k_B T}{2} \log\left[(1 + J^{(+)}I^{(+)})(1 + J^{(-)}I^{(-)}) - J^{(+)}J^{(-)}I^2(1)\right], \tag{67}$$

where we introduced the abbreviations $J^{(\pm)} = J(3/2 \pm \nu)$ and $I^{(\pm)} = I(3/2 \pm \nu)$. For the further analysis, we now express the series (66) in terms of the generalized Lambert series (57) as

$$I(c) = \mathcal{L}_Z(0, c) - \mathcal{L}_{Z^2}(0, c). \tag{68}$$

Applying the asymptotic expansion of the generalized Lambert series [31], we obtain

$$I(c) = \frac{1}{2\mu}\left[-\log\left(\frac{\mu}{2}\right) - \psi(c) + \sum_{n=1}^{\infty} \frac{B_{2n}(2^{2n} - 2)}{2n(2n)!}B_{2n}(c)\mu^{2n}\right], \tag{69}$$

with the digamma function $\psi(c)$ and the Bernoulli polynomial $B_{2n}(c)$ [26]. For $c = 1$, the functions $\psi(c)$ and $B_{2n}(c)$ are given by the negative Euler-Mascheroni constant, $-\gamma = -0.57721\ldots$ and the Bernoulli numbers $B_{2n}$, respectively.

Before proceeding with our calculation, we note that a short-distance expansion of the capacitance coefficients for two general spheres has already been carried out in [32] and [33], where the latter one also applied the asymptotic expansion of the generalized Lambert series. Besides using a different definition of the dimensionless capacitance coefficients and geometric parameters, we determine, in contrast to previous work, a complete expansion of the functions $I(c)$ as well as $J(c)$ in powers of $\mu$. In the definition of our geometric parameters, we follow the notation common in the Casimir community which also simplifies to obtain the limits of equal spheres and of the sphere-plane geometry.

In order to obtain a complete expansion of the argument of the logarithm in (67) in powers of $\mu$, we need to account for the fact that $I^{(\pm)}$ and $J^{(\pm)}$ depend on $\mu$ through $c = 3/2 \pm \nu(\mu)$. By making use of the Taylor series for $\nu$ given in (62) with the coefficients (63) and (64), one immediately obtains a corresponding Taylor series for $c(\mu)$ which is required to determine the Taylor series in powers of $\mu$ for the digamma function $\psi(c(\mu))$ and the Bernoulli polynomial $B_{2n}(c(\mu))$ appearing in (69).

For $\mu \leq 1$, a condition fulfilled in the small-distance limit, and with the help of (27), the digamma function can be expanded as

$$\psi(c(\mu)) = \psi(c_0) + \sum_{m=1}^{\infty} \sum_{n=1}^{m} \frac{\psi^{(n)}(c_0)}{n!} \hat{B}_{m,n}(c_1, c_2, \ldots)\mu^{2m}, \tag{70}$$

where $\psi^{(n)}(c_0)$ denotes the polygamma function [26] and $\hat{B}_{m,n}(c_1, c_2, \ldots)$ are partial ordinary Bell polynomials. Correspondingly, the expansion of the Bernoulli polynomials yields

$$B_{2n}(c(\mu)) = \frac{1}{\mu^{2n}} \sum_{k=0}^{2n} \binom{2n}{k} B_k \sum_{m=n}^{\infty} \hat{B}_{m+n-k,2n-k}(c_0, c_1, \ldots)\mu^{2m}. \tag{71}$$

Note that the arguments of the Bell polynomials in (70) and (71) differ.

Inserting the expansions from above into (69), we find the series expansion

$$I(c) = \frac{1}{2\mu} \sum_{m=0}^{\infty} I_m(c)\mu^{2m}, \tag{72}$$

with the coefficients

$$I_0(c) = -\log(\mu/2) - \psi(c_0), \tag{73}$$

and for $m > 0$

$$I_m(c) = \sum_{n=1}^{m} \left[ \frac{B_{2n}(2^{2n}-2)}{2n} \sum_{k=-n}^{n} \frac{B_{n+k}\hat{B}_{m-k,n-k}(c_0,c_1,\dots)}{(n-k)!(n+k)!} - \frac{\psi^{(n)}(c_0)}{n!}\hat{B}_{m,n}(c_1,c_2,\dots) \right]. \tag{74}$$

The prefactor (65) can be expanded correspondingly and we obtain

$$J(c) = 2\mu \sum_{m=0}^{\infty} J_m(c)\mu^{2m}, \tag{75}$$

with the coefficients

$$J_m(c) = \sum_{n=0}^{m} \frac{\hat{B}_{m+n+1,2n+1}(c_0-1,c_1,c_2,\dots)}{(2n+1)!}. \tag{76}$$

By means of (72) and (76) one can derive a systematic expansion of the monopole contributions (67) for small distances. The optimal cut-off for this asymptotic series is discussed in Ref. [33]. However, even the calculation of the terms up to order $\mu^4$ involves a decent amount of algebra which we relegate to Appendix A. Expanding the result (103) in a Mercator series finally yields

$$\Delta \approx \frac{k_B T}{2} \left\{ \log\left[\epsilon_0(\gamma - \log(\mu/2)) + \delta_0\right] + \frac{1}{6}\frac{\epsilon_1(\gamma - \log(\mu/2)) + \delta_1}{\epsilon_0(\gamma - \log(\mu/2)) + \delta_0}\mu^2 \right.$$
$$\left. + \frac{1}{360}\left[ \frac{3\left[\epsilon_2(\gamma - \log(\mu/2)) + \delta_2\right]}{\epsilon_0(\gamma - \log(\mu/2)) + \delta_0} - \frac{5\left[\epsilon_1(\gamma - \log(\mu/2)) + \delta_1\right]^2}{\left[\epsilon_0(\gamma - \log(\mu/2)) + \delta_0\right]^2} \right]\mu^4 \right\}. \tag{77}$$

The expansion coefficients $\epsilon_n(u)$ and $\delta_n(u)$ are defined in (104)–(109) in Appendix A and depend only on the geometric parameter

$$u = \frac{R_{\text{eff}}^2}{R_1 R_2}. \tag{78}$$

This parameter can take arbitrary values between 0 and 1/4, corresponding to the sphere-plane geometry and equally sized spheres, respectively. Table 1 gives the values of the expansion coefficients for the two limiting cases. The sphere-plane limit, $u = 0$, is consistent with the results given in [5]. Their numerical constants $\gamma_i$, $i = 1, 2, 3, 4$ can now be expressed analytically as

$$\gamma_1 = \gamma + \log(2) \tag{79}$$

$$\gamma_2 = \gamma_1 + \frac{1}{12} \tag{80}$$

$$\gamma_3 = \frac{1}{2}\left(5\gamma_2^2 - 3\gamma_1^2 - \frac{107}{120}\gamma_1\right) \tag{81}$$

$$\gamma_4 = 5\gamma_2 - 3\gamma_1 - \frac{107}{240}. \tag{82}$$

Table 1: Expansion coefficients $\epsilon_n(u)$ and $\delta_n(u)$ appearing in (77) in the limits of the sphere-plane geometry ($u = 0$) and of equal spheres ($u = 1/4$).

| $u$ | $0$ | $1/4$ |
|---|---|---|
| $\epsilon_0(u)$ | $1$ | $\log(2)$ |
| $\delta_0(u)$ | $0$ | $\log^2(2)$ |
| $\epsilon_1(u)$ | $1$ | $\frac{1}{2}\left(\log(2) - \frac{1}{8}\right)$ |
| $\delta_1(u)$ | $\frac{1}{12}$ | $\frac{1}{2}\left(\log^2(2) - \frac{1}{12}\log(2)\right)$ |
| $\epsilon_2(u)$ | $1$ | $\frac{1}{3}\left(\log(2) - \frac{47}{128}\right)$ |
| $\delta_2(u)$ | $\frac{107}{360}$ | $\frac{1}{3}\left(\log^2(2) - \frac{83}{320}\log(2) - \frac{5}{384}\right)$ |

## 6 Conclusions

We have for the first time derived an exact analytical expression for the Casimir free energy of two Drude spheres of arbitrary radii completely within the scattering approach common in Casimir physics. In contrast to previous work on the sphere-plane geometry and two spheres of equal radii, the plane-wave basis was used, which led to a connection with a combinatorial problem. The structure of this combinatorial problem highlights the difference between the general two-sphere case and the corresponding limiting cases. The scattering approach also provides an intuitive interpretation of the structure of the result for the Casimir free energy.

Earlier work by Fosco et al. [12] has pointed out the relevance of the capacitance matrix for the Casimir free energy in the high-temperature limit. However, the fact that an analytical expression for the capacitance matrix exists even for two spheres of different radii seems to have largely escaped the attention of the Casimir community. Our work thus provides an interesting connection between Casimir physics and electrostatics. This is in particular the case for the short-distance expansion where by profiting from results obtained within the electrostatics community, we derived a systematic expansion in powers of $\mu$ which might be useful in that community as well.

## Acknowledgements

The authors are grateful to Michael Hartmann, Astrid Lambrecht, Paulo Maia Neto, Serge Reynaud and Benjamin Spreng for many inspiring discussions. Benjamin Spreng has also kindly provided numerical data for comparison between analytical and numerical results in the general sphere-sphere geometry.

## A  Coefficients of the short-distance expansion

In this appendix, we derive expressions for the coefficients $\epsilon_0, \delta_0, \epsilon_1, \delta_1, \epsilon_2,$ and $\delta_2$ appearing in the short-distance expansion (77). While the expansion parameter $\mu$ depends on the distance between the two spheres, the geometric parameter $u$ defined in (78) is a function of the sphere radii alone. The parameter $v_0$ introduced in (63) can be expressed in terms of $u$ as

$$v_0 = \text{sgn}(R_1 - R_2)\frac{\sqrt{1 - 4u}}{2}. \tag{83}$$

We remark that the sign of the difference of radii will not show up in the final expressions. As we will start from the results (72)–(76), we need to express the coefficients $c_n$ in terms of the

Taylor coefficients $v(\mu)$. From $c = 3/2 \pm v$ and together with (83) we obtain for the first three coefficients

$$c_0 = \frac{3}{2} \pm v_0 , \tag{84}$$

$$c_1 = \mp \frac{v_0}{3} u , \tag{85}$$

$$c_2 = \mp \frac{v_0}{60} u(1 - 12u) . \tag{86}$$

For convenience of the reader, we list the partial ordinary Bell polynomials for the index combinations needed in the following:

$$\hat{B}_{n,0}(x_1, x_2, \ldots) = \delta_{n,0} , \tag{87}$$

$$\hat{B}_{n,1}(x_1, x_2, \ldots) = x_n , \tag{88}$$

$$\hat{B}_{n,n-1}(x_1, x_2, \ldots) = (n-1)x_1^{n-2}x_2 , \tag{89}$$

$$\hat{B}_{n,n}(x_1, x_2, \ldots) = x_1^n . \tag{90}$$

For the monopole contributions (67) to order $\mu^4$ we need the coefficients of $I(1)$, $I^{(\pm)}$, and $J^{(\pm)}$ up to second order. After some tedious but straightforward algebra, we obtain from (69)

$$I_0(1) = \gamma - \log\left(\frac{\mu}{2}\right) , \tag{91}$$

$$I_1(1) = \frac{1}{72} , \tag{92}$$

$$I_2(1) = \frac{7}{43200} . \tag{93}$$

From (73) and (74), we find

$$I_0^{(\pm)} = \gamma - \log\left(\frac{\mu}{2}\right) - \Psi_0^{(\pm)} , \tag{94}$$

$$I_1^{(\pm)} = \frac{1}{72} - \frac{u}{12} + \frac{1}{6}\left(\frac{1}{2} \pm v_0\right) - \Psi_1^{(\pm)} , \tag{95}$$

$$I_2^{(\pm)} = \frac{7}{43200} + \frac{48u + 73u^2}{1440} - \frac{7 + 13u}{360}\left(\frac{1}{2} \pm v_0\right) - \Psi_2^{(\pm)} , \tag{96}$$

where we introduced

$$\Psi_0^{(\pm)} = \psi\left(\frac{3}{2} \pm v_0\right) + \gamma , \tag{97}$$

$$\Psi_1^{(\pm)} = \pm v_1 \psi^{(1)}\left(\frac{3}{2} \pm v_0\right) , \tag{98}$$

$$\Psi_2^{(\pm)} = \pm v_2 \psi^{(1)}\left(\frac{3}{2} \pm v_0\right) + \frac{v_1^2}{2}\psi^{(2)}\left(\frac{3}{2} \pm v_0\right) . \tag{99}$$

Including the Euler-Mascheroni constant in $\Psi_0^{(\pm)}$ will help to simplify the final expressions. We note that the coefficients (91)–(93) can be obtained from (94)–(96) by choosing the lower sign and setting $v_0 = 1/2$, i.e. $u = 0$.

From (76), we finally obtain

$$J_0^{(\pm)} = \frac{1}{2} \pm v_0 , \tag{100}$$

$$J_1^{(\pm)} = \frac{1 - 3u}{6}\left(\frac{1}{2} \pm v_0\right) , \tag{101}$$

$$J_2^{(\pm)} = \frac{1 - 15u(1 - 3u)}{120}\left(\frac{1}{2} \pm v_0\right) . \tag{102}$$

We now insert the coefficients just derived into the monopole contributions (67) and sort the product terms by powers of $\mu$. The result can be written as

$$\Delta = \frac{k_B T}{2} \log\left\{ \sum_{n=0}^{2} \frac{\mu^{2n}}{(2n+1)!} \left[ \epsilon_n(u)\left(\gamma - \log\frac{\mu}{2}\right) + \delta_n(u) \right] + \mathcal{O}(\mu^6) \right\}, \tag{103}$$

with the coefficients

$$\epsilon_0(u) = 1 - u\varphi_{0,0} \tag{104}$$

$$\delta_0(u) = 1 - \varphi_{0,1} + u\theta_{0,0} \tag{105}$$

$$\epsilon_1(u) = 1 - 2u - u^2 - 2u(1-3u)\varphi_{0,0} - 6u\varphi_{1,0} \tag{106}$$

$$\delta_1(u) = \frac{1}{12}(13-30u) - \frac{1}{12}u(13-6u)\varphi_{0,0} - (1-4u)\varphi_{0,1} - 6\varphi_{1,1} \tag{107}$$
$$+ 2u(1-3u)\theta_{0,0} + 6u(\theta_{0,1} + \theta_{1,0})$$

$$\epsilon_2(u) = \frac{1}{6}(6 - 64u + 132u^2 + 193u^3) - \frac{2}{3}u(8 - 75u + 180u^2)\varphi_{0,0} \tag{108}$$
$$- 40u(1-3u)\varphi_{1,0} - 120u\varphi_{2,0}$$

$$\delta_2(u) = \frac{1}{360}(467 - 5240u + 14810u^2 + 300u^3) \tag{109}$$
$$- \frac{1}{120}u(589 - 3040u + 1930u^2)\varphi_{0,0}$$
$$- \frac{1}{3}(3 - 58u + 208u^2)\varphi_{0,1}$$
$$- \frac{5}{3}u(13-6u)\varphi_{1,0} - 20(1-4u)\varphi_{1,1} - 120\varphi_{2,1}$$
$$+ \frac{2}{3}u(8 - 75u + 180u^2)\theta_{0,0}$$
$$+ 40u(1-3u)(\theta_{0,1} + \theta_{1,0}) + 120u\theta_{1,1} + 120u(\theta_{0,2} + \theta_{2,0}).$$

Here, we have introduced abbreviations for the sums of the functions (97)–(99)

$$\varphi_{n,m} = \left(\frac{1}{2} + v_0\right)^m \Psi_n^{(+)} + \left(\frac{1}{2} - v_0\right)^m \Psi_n^{(-)} \tag{110}$$

as well as for their products

$$\theta_{n,m} = \Psi_n^{(+)}\Psi_m^{(-)}. \tag{111}$$

For the limiting cases of the sphere-plane geometry ($u = 0$) and two spheres of equal radii ($u = 1/4$), the coefficients $v_1$ and $v_2$ vanish. Hence, $\varphi_{1,m}$ and $\varphi_{2,m}$ yield zero and all $\theta_{n,m}$ except for $\theta_{0,0}$ vanish. The expressions for $\epsilon_n$ and $\delta_n$ then simplify to the results listed in Table 1.

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
