# Peer review of "Classical Casimir free energy for two Drude spheres of arbitrary radii: A plane-wave approach"

_SciPost Physics, doi:SciPost Phys. Core 4, 011 (2021)_

## Round 1 · Referee Report · Anonymous (Referee 1) · 2020-11-30

Strengths

  1. The paper contains a detailed analytical calculation of the Casimir free-energy for the geometry of two Drude spheres of arbitrary radii, in the high-temperature limit.
  2. An interesting connection between Casimir physics and electrostatics is highlighted.

Weaknesses

  1. The main result can be obtained from existent results in the literature, so the complex calculations presented in the paper could be avoided. There is no clear justification for using an alternative approach based on the scattering formalism with a plane-wave basis.

Report

In this paper, the authors compute the Casimir free energy for two Drude spheres of arbitrary radii, in the high-temperature limit, using the scattering approach with a plane-wave basis. The main results are given by Eq. (15), which gives the free energy for Dirichlet spheres, and Eq.(40), which gives the difference between the free energy for Drude and Dirichlet boundary conditions. The combination of both results gives an analytical expression for the Casimir free energy for Drude spheres, which is shown to reproduce correctly the known cases of a sphere in front of a plane, and two spheres of equal radii. Moreover, it reproduces the proximity force approximation in the short-distance expansion. The paper is clearly written and describes a technically complex calculation.
My main concern is the following: Eq. (15) has been previously obtained in Ref.[5], while Eq.(40) is a particular case of a general result obtained in Ref.[13]. If the main aim of the authors is to obtain an analytical result for the sphere-sphere Casimir free energy, many calculations could be avoided. If, instead, their main goal is to reproduce previous results using an alternative method, they should state this from the very beginning, and provide motivations to do that.

Requested changes

  1. In the Introduction, the authors should mention that although the analytical formula for Drude spheres of different radii could be obtained from the results of Refs. [5] and [13], they will obtain the formula using an alternative approach, providing the motivations for doing that.
  2. In Section 4.4, the relation between Eq.(40) and the general result of Ref.[13] should be clearly stated. The quantity Δ is formally evaluated in [13] for a system on N conductors of arbitrary shapes, and involves the logarithm of the determinant of the capacitance matrix. When evaluated for two spheres, this formula, which could be written in the paper, reproduces Eq.(40), up to terms not relevant for the evaluation of the force. I think this would reinforce the interesting connection between Casimir physics and electrostatics mentioned by the authors in the Conclusions.

  • validity: top
  • significance: ok
  • originality: good
  • clarity: high
  • formatting: excellent
  • grammar: excellent

Author:  Gert-Ludwig Ingold  on 2021-02-19  [id 1252]

(in reply to Report 1 on 2020-11-30)

We thank the referee for the valuable comments and suggestions which lead us to modify the manuscript for resubmission as indicated below.

Firstly though, we would like to comment on the weakness listed by the referee. While this point is certainly valid, we see a number of reasons why we think that our paper deserves publication.

  • While, as the referee points out and as we have stated already in the first version of the manuscript, a combination of results in the literature allows to obtain the main result of our paper, it has, to the best of our knowledge, never been stated explicitly. As we had mentioned already in the first version of the manuscript, statements in the literature indicate that the existence of an analytical result for spheres with different radii was not known in the Casimir community. Discussions with colleagues have supported this view and we ourselves became aware of it only while finalising the manuscript. As we discussed already in the first version, the high-temperature result is relevant to the analysis of experiments even at lower temperatures and therefore, we believe that it is important to explicitly state the result (51) for the free energy.
  • While our result could be obtained by combining results from three different sources (refs. 5, 12, and 16), as far as we know it is the first time that the Casimir free energy for two Drude spheres of different radii is derived within a single framework, in our case within the scattering approach with a plane-wave basis.
  • The derivation of our main result may appear to be quite involved and it is for this reason that we tried to be sufficiently explicit for the reader to follow without too many difficulties. We like to highlight two more formal points, though. Our calculation connects the scattering between two different objects with an interesting combinatorial problem which is solved on the way to our result (51). Furthermore, we expect that this approach might also be useful in other problems related to the Casimir effect.
  • For completeness, we mention the connection between Casimir physics and electrostatics which our calculation highlights, a point which was also acknowledged by the referee as a strength of our paper.

We now specifically discuss the referee's suggestions and how we addressed them.

1. In the Introduction, the authors should mention that although the analytical formula for Drude spheres of different radii could be obtained from the results of Refs. [5] and [13], they will obtain the formula using an alternative approach, providing the motivations for doing that.

Reply:

We have extended the introduction to explain in more detail the motivation of our work. Specifically, we have modified the last part of the fourth paragraph (previously the third paragraph) discussing the semi-analytical approach by Bimonte (ref. 6) and the relevance of our work in this respect. We then have modified the third paragraph and added two more paragraphs motivating our work before discussing the structure of the paper.

2. In Section 4.4, the relation between Eq. (40) and the general result of Ref. [13] should be clearly stated. The quantity Δ is formally evaluated in [13] for a system on N conductors of arbitrary shapes, and involves the logarithm of the determinant of the capacitance matrix. When evaluated for two spheres, this formula, which could be written in the paper, reproduces Eq.(40), up to terms not relevant for the evaluation of the force. I think this would reinforce the interesting connection between Casimir physics and electrostatics mentioned by the authors in the Conclusions.

Reply:

We have followed the advice and extended the discussion in the first part of section 4.4 in order to make the connection between our result (41) (eq. 40 in the first version) and the paper by Fosco et al. [12] (ref. 13 in the first version) even more explicit. Moreover, after (49), we now emphasize the connection between the capacitance coefficients in [16] and our round-trip description. Furthermore, we added the historic reference [27].

---

## Round 1 · Referee Report · Anonymous (Referee 2) · 2020-12-22

Strengths

1 The result appears to be a significant generalisation of for the thermal Casimir interaction between two Drude spheres of differing radii.

2 The calculation is very involved but the final result is quite elegant.

3 The paper recovers previously known results as special cases

Weaknesses

1 The paper is aimed at a very narrow audience, not even at the general Casimir community

2 The presentation should be improved

Report

The authors compute the thermal component of the Casimir free energy between two Drude spheres of unequal radii using the scattering approach. They carry out their calculation in the plane wave basis and find a formula which agrees with the results for the sphere plane set up and for two spheres of equal radii. The paper is very technical and some aspects should be made clearer to make the paper accessible to a more general readership. The authors use a method based on correcting the problem of a scaler field with Dirichlet boundary conditions to obtain the solution for the Drude problem. The difference between the elements in the expansion for the free energy are seen clearly between equations (8) and (9). However the computation is not straightforward and requires considerable combinatorial analysis. I think that the results of the paper are interesting. Although the calculation is very long to check it agrees with known results in certain established limits and the authors claim to have tested it numerically (although this is not shown explicitly). The paper thus seems suitable for publication.

Requested changes

1 The general level of English could be improved and the introductory part of the paper needs some improvement as it stands it needs a bit more explanation. See the points below.

2 P1 However, also thermal photons contribute to the Casimir force which survives the classical limit This could be made clearer eg Thermal photons also contribute to the Casimir force. The authors could perhaps say here that the zero frequency Matsubara term is non zero and yields the thermal component of the Casimir force which is nonzero.

3 P1 Then, the free energy does no longer depend on Planck’s constant and is found to be linear in temperature - should be rewritten given the comments made about the previous sentence

4 P1 Here, the terms for non-zero Matsubara frequencies are treated within the derivative expansion while this approximation is less accurate for zero frequency. An exact analytical high-temperature expression will thus be valuable. These two phrases are confused/not clear

5 P1 … of a scalar field which is found to be dual to the known result .. what do the authors mean by dual. The sentence is too concise to convey any meaning so it should either be expanded on or discussed later. Later on we see that the authors rederive the result of [5] but their result is given in a different form. I would thus say their result is equivalent to that of [5] in this case.

6 P3 Within the scattering approach to the Casimir effect [2], the free energy can be expressed as a Matsubara sum - this is a bit misleading, in any approach the free energy is expressed this way, and it is more common to say a sum over Matsubara frequencies.

7 P3 It would help the reader to give the definition of a Drude type metal where it is mentioned - I think this should be discussed in a detailed way as it is a crucial point in the paper.

8 P5 Making use of the symmetries of cosine and hyperbolic cosine - I am not sure what is meant by this

9 P5 again the same mysterious sentence - Casimir free energy which is found to be dual to the known result [5].

10 P5 After having determined the matrices associated with the bilinear forms in the exponentials, our main task will be to evaluate the corresponding determinants - I guess he authors mean the integrals in equation 8 which can be written as a determinant ?

11 P7 because the sum is converging considerably faster - should be the sum converges faster (I am not sure considerably should be used without more justification).

12 P7 after eq 19 it would be helpful to explain the difference in boundary conditions for the non expert reader if it has not already be done in response to the point above.

13 P7 To obtain the result for the electromagnetic case from the scalar case, we need to determine the contribution of all terms which contain at least one factor −1 when the product in (8) is expanded - I see what this means but it should expressed more precisely.

  • validity: high
  • significance: high
  • originality: good
  • clarity: ok
  • formatting: excellent
  • grammar: reasonable

Author:  Gert-Ludwig Ingold  on 2021-02-19  [id 1253]

(in reply to Report 2 on 2020-12-22)

First, we would like to thank the referee for the detailed and constructive report. We are pleased that the referee finds our paper in principle suitable for publication.

Before detailing the changes for resubmission in response to the referee's suggestions, thereby addressing point 2 of the listed weaknesses, we would like to comment on point 1 where the referee points out that our result is only aimed at a narrow audience.

While the length of the paper and the more formal character of parts of the paper might indeed indicate that we address a narrow audience, we do not believe this to be the case for the following reasons.

  • The explicit result (51) for the Casimir free energy, from which the Casimir force can be obtained, is not only relevant to theorists but also to experimentalists while analysing Casimir experiments on two spheres with different radii. We have rewritten the second part of the fourth paragraph (previously the third paragraph) of the introduction to make this point clearer.
  • While the plane-sphere geometry still is the most relevant geometry for Casimir experiments, experiments on the sphere-sphere geometry have been carried out lately. In particular, one of these experiments (ref. 7) addresses larger distances, thereby rendering the contribution of thermal photons and the zero-frequency Matsubara term more relevant. Furthermore, by changing the salt concentration of the aequous medium, ref. 7 is able to modify the strength of precisely the zero-frequency part. We have modified the third paragraph to mention this point.
  • Beyond the Casimir community in a more narrow sense, the sphere-sphere geometry is relevant for the colloid community. To emphasize this point, we have modified the first part of the third paragraph of the introduction by splitting the earlier list of three experimental references into a part pertaining to the Casimir effect (now refs. 7 and 8) and another part related to colloidal systems (now ref. 9 and a new ref. 10). In colloidal systems, the comment in the last two sentences of our previous point also applies in principle.

Therefore, we are convinced that our results are of interest not only to a small group of theorists interested in a specific aspect of the Casimir effect but in fact to a larger community.

As suggested by the referee, we have worked on the overall presentation and in particular, addressed all points raised by the referee as detailed below. In a couple of cases, we did not follow the referee for reasons given below as well. We hope that the changes adequately address the referee's criticism.

2 P1 However, also thermal photons contribute to the Casimir force which survives the classical limit. This could be made clearer eg Thermal photons also contribute to the Casimir force. The authors could perhaps say here that the zero frequency Matsubara term is non zero and yields the thermal component of the Casimir force which is nonzero.

Reply:

We significantly revised the first paragraph of our introduction to make the role of thermal photons for the Casimir interaction clearer. In particular, we discuss the equivalence of the classical limit (ℏ→0) and the high-temperature limit (T→∞).

3 P1 Then, the free energy does no longer depend on Planck’s constant and is found to be linear in temperature - should be rewritten given the comments made about the previous sentence

Reply:

We addressed this point by the changes discussed in the previous point.

4 P1 Here, the terms for non-zero Matsubara frequencies are treated within the derivative expansion while this approximation is less accurate for zero frequency. An exact analytical high-temperature expression will thus be valuable. These two phrases are confused/not clear

Reply:

We wanted to convey, that the semi-analytical approach by Bimonte uses the derivative expansion to determine the contributions from the non-zero Matsubara frequencies. However, for the zero-frequency term, it is common to use the exact analytical expression. The expressions are known for a sphere-plane geometry and a geometry of two equal spheres. Our generalization of the classical result for a system of spheres with arbitrary radii might therefore also be helpful for such semi-analytical approaches.

We hope that our revision of the relevant part of the introduction (the last part of the fourth paragraph) clarifies our argument.

5 P1 … of a scalar field which is found to be dual to the known result .. what do the authors mean by dual. The sentence is too concise to convey any meaning so it should either be expanded on or discussed later. Later on we see that the authors rederive the result of [5] but their result is given in a different form. I would thus say their result is equivalent to that of [5] in this case.

Reply:

Our result is dual, in the sense that the known expression [5] is expressed in terms of bispherical multipoles. However, our approach is based on a round-trip description. Using 'dual' instead of 'equivalent', we wanted to emphasise that our result is not only equivalent to the known one but also allows for an alternative interpretation in terms of round trips.

We have modified the sentence referring to section 3 in the last paragraph of the introduction accordingly. More details concerning this duality are then given later in the paper.

6 P3 Within the scattering approach to the Casimir effect [2], the free energy can be expressed as a Matsubara sum - this is a bit misleading, in any approach the free energy is expressed this way, and it is more common to say a sum over Matsubara frequencies.

Reply:

While we disagree that the free energy is always expressed as a sum over Matsubara frequencies - one could also integrate over real frequencies - we followed the suggestion of the referee and modified the text to avoid using the term 'Matsubara sum'.

7 P3 It would help the reader to give the definition of a Drude type metal where it is mentioned - I think this should be discussed in a detailed way as it is a crucial point in the paper.

Reply:

We extended the first paragraph of section 2. Specifically, we now introduce the dielectric function for a Drude metal and point out that the resulting finite dc conductivity leads to a simplification concerning the modes contributing to the Casimir free energy.

8 P5 Making use of the symmetries of cosine and hyperbolic cosine - I am not sure what is meant by this

Reply:

We were referring to the fact that adding π to the argument of the cosine will merely change the sign and that the hyperbolic cosine is an even function. These properties are of course well known and therefore we refrain from such a detailed discussion in the paper. Since our original hint at the symmetries apparently can constitute a source of confusion, we removed this sentence.

9 P5 again the same mysterious sentence - Casimir free energy which is found to be dual to the known result [5].

Reply:

While we added an explanation of the meaning of duality in the introduction, we have replaced ‘dual’ by ‘equivalent’ in this case.

10 P5 After having determined the matrices associated with the bilinear forms in the exponentials, our main task will be to evaluate the corresponding determinants - I guess the authors mean the integrals in equation 8 which can be written as a determinant ?

Reply:

We do not think that it is necessary to amend the discussion following (9) (previously eq. 8). We trust that the reader knows how to evaluate Gaussian integrals. However, we want to emphasize that the non-trivial step in doing these integrals is to evaluate the determinant of the bilinear form in the exponent.

11 P7 because the sum is converging considerably faster - should be the sum converges faster (I am not sure considerably should be used without more justification).

Reply:

We agree with the statement in parentheses. Since a detailed discussion of convergence properties is beside the main point of the paper, we removed the term ‘considerably’.

12 P7 after eq 19 it would be helpful to explain the difference in boundary conditions for the non expert reader if it has not already be done in response to the point above.

Reply:

In our opinion, the additional explanations at the beginning of section 2 added in response to the referee's point 7 are sufficient to understand the boundary conditions.

13 P7 To obtain the result for the electromagnetic case from the scalar case, we need to determine the contribution of all terms which contain at least one factor −1 when the product in (8) is expanded - I see what this means but it should express more precisely.

Reply:

We have extended the discussion at the beginning of section 4.1 accordingly. We now recall the meaning of the -1 term in (9), previously eq. 8, to clarify how each term accounting for at least one factor -1 in (9), contributes to the monopole part for Dirichlet spheres.

---

## Round 2 · Referee Report · Anonymous (Referee 2) · 2021-2-26

Report

The authors have revised the paper and modified the manuscript in a completely satisfactory manner as far as I am concerned and the manuscript is I believe suitable for publication.

---

## Round 2 · Referee Report · Anonymous (Referee 1) · 2021-4-1

Strengths

I maintain the previous ones: 1. The paper contains a detailed analytical calculation of the Casimir free-energy for the geometry of two Drude spheres of arbitrary radii, in the high-temperature limit. 2. An interesting connection between Casimir physics and electrostatics is highlighted.

Weaknesses

  1. The calculations are rather complex and difficult to follow.

Report

The relation with previous works has been clarified. In my opinion, the paper meets the acceptance criteria of the journal. I recommend publication in its present form.

Requested changes

No further suggestions.

---

## Round 2 · Author Response

Dear editor,

we would like to thank you and the referees for assessing our manuscript. We have prepared a revised version of the manuscript which accounts for the points raised by the referees as discussed in detail below. We believe that the feedback has helped us to further improve the manuscript.

In the discussion below, we have also addressed the weak points noted by the two referees by stating a number of counter-arguments. We hope that in view of these arguments and the amendments, our manuscript will now be found suitable for publication in SciPost.

Best regards, Tanja Schoger, Gert-Ludwig Ingold

Reply to referee 1

We thank the referee for the valuable comments and suggestions which have led us to modify the manuscript as detailed below.

Firstly though, we would like to comment on the weakness listed by the referee. While this point is certainly valid, we see a number of reasons why we think that our paper deserves publication.

  • While, as the referee points out and as we have stated already in the first version of the manuscript, a combination of results in the literature allows to obtain the main result of our paper, it has, to the best of our knowledge, never been stated explicitly. As we had mentioned already in the first version of the manuscript, statements in the literature indicate that the existence of an analytical result for spheres with different radii was not known in the Casimir community. Discussions with colleagues have supported this view and we ourselves became aware of it only while finalising the manuscript. As we discussed already in the first version, the high-temperature result is relevant to the analysis of experiments even at lower temperatures and therefore, we believe that it is important to explicitly state the result (51) for the free energy.

  • While our result could be obtained by combining results from three different sources (refs. 5, 12, and 16), as far as we know it is the first time that the Casimir free energy for two Drude spheres of different radii is derived within a single framework, in our case within the scattering approach with a plane-wave basis.

  • The derivation of our main result may appear to be quite involved and it is for this reason that we tried to be sufficiently explicit for the reader to follow without too many difficulties. We like to highlight two more formal points, though. Our calculation connects the scattering between two different objects with an interesting combinatorial problem which is solved on the way to our result (51). Furthermore, we expect that this approach might also be useful in other problems related to the Casimir effect.

  • For completeness, we mention the connection between Casimir physics and electrostatics which our calculation highlights, a point which was also acknowledged by the referee as a strength of our paper.

We now specifically discuss the referee's suggestions and how we addressed them.

  1. In the Introduction, the authors should mention that although the analytical formula for Drude spheres of different radii could be obtained from the results of Refs. [5] and [13], they will obtain the formula using an alternative approach, providing the motivations for doing that.

    Reply:

    We have extended the introduction to explain in more detail the motivation of our work. Specifically, we have modified the last part of the fourth paragraph (previously the third paragraph) discussing the semi-analytical approach by Bimonte (ref. 6) and the relevance of our work in this respect. We then have modified the third paragraph and added two more paragraphs motivating our work before discussing the structure of the paper.

  2. In Section 4.4, the relation between Eq. (40) and the general result of Ref. [13] should be clearly stated. The quantity Δ is formally evaluated in [13] for a system on N conductors of arbitrary shapes, and involves the logarithm of the determinant of the capacitance matrix. When evaluated for two spheres, this formula, which could be written in the paper, reproduces Eq.(40), up to terms not relevant for the evaluation of the force. I think this would reinforce the interesting connection between Casimir physics and electrostatics mentioned by the authors in the Conclusions.

    Reply:

    We have followed the advice and extended the discussion in the first part of section 4.4 in order to make the connection between our result (41) (eq. 40 in the first version) and the paper by Fosco et al. [12] (ref. 13 in the first version) even more explicit. Moreover, after (49), we now emphasize the connection between the capacitance coefficients in [16] and our round-trip description. Furthermore, we added the historic reference [27].

Reply to referee 2

First, we would like to thank the referee for the detailed and constructive report. We are pleased that the referee finds our paper in principle suitable for publication.

Before detailing the changes, we have made in response to the referee's suggestions, thereby addressing point 2 of the listed weaknesses, we would like to comment on point 1 where the referee points out that our result is only aimed at a narrow audience.

While the length of the paper and the more formal character of parts of the paper might indeed indicate that we address a narrow audience, we do not believe this to be the case for the following reasons.

  • The explicit result (51) for the Casimir free energy, from which the Casimir force can be obtained, is not only relevant to theorists but also to experimentalists while analysing Casimir experiments on two spheres with different radii. We have rewritten the second part of the fourth paragraph (previously the third paragraph) of the introduction to make this point clearer.

  • While the plane-sphere geometry still is the most relevant geometry for Casimir experiments, experiments on the sphere-sphere geometry have been carried out lately. In particular, one of these experiments (ref. 7) addresses larger distances, thereby rendering the contribution of thermal photons and the zero-frequency Matsubara term more relevant. Furthermore, by changing the salt concentration of the aequous medium, ref. 7 is able to modify the strength of precisely the zero-frequency part. We have modified the third paragraph to mention this point.

  • Beyond the Casimir community in a more narrow sense, the sphere-sphere geometry is relevant for the colloid community. To emphasize this point, we have modified the first part of the third paragraph of the introduction by splitting the earlier list of three experimental references into a part pertaining to the Casimir effect (now refs. 7 and 8) and another part related to colloidal systems (now ref. 9 and a new ref. 10). In colloidal systems, the comment in the last two sentences of our previous point also applies in principle.

Therefore, we are convinced that our results are of interest not only to a small group of theorists interested in a specific aspect of the Casimir effect but in fact to a larger community.

As suggested by the referee, we have worked on the overall presentation and in particular, addressed all points raised by the referee as detailed below. In a couple of cases, we did not follow the referee for reasons given below as well. We hope that the changes adequately address the referee's criticism.

2 P1 However, also thermal photons contribute to the Casimir force which survives the classical limit. This could be made clearer eg Thermal photons also contribute to the Casimir force. The authors could perhaps say here that the zero frequency Matsubara term is non zero and yields the thermal component of the Casimir force which is nonzero.

Reply:

We significantly revised the first paragraph of our introduction to make the role of thermal photons for the Casimir interaction clearer. In particular, we discuss the equivalence of the classical limit (ℏ→0) and the high-temperature limit (T→∞).

3 P1 Then, the free energy does no longer depend on Planck’s constant and is found to be linear in temperature - should be rewritten given the comments made about the previous sentence

Reply:

We addressed this point by the changes discussed in the previous point.

4 P1 Here, the terms for non-zero Matsubara frequencies are treated within the derivative expansion while this approximation is less accurate for zero frequency. An exact analytical high-temperature expression will thus be valuable. These two phrases are confused/not clear

Reply:

We wanted to convey, that the semi-analytical approach by Bimonte uses the derivative expansion to determine the contributions from the non-zero Matsubara frequencies. However, for the zero-frequency term, it is common to use the exact analytical expression. The expressions are known for a sphere-plane geometry and a geometry of two equal spheres. Our generalization of the classical result for a system of spheres with arbitrary radii might therefore also be helpful for such semi-analytical approaches.

We hope that our revision of the relevant part of the introduction (the last part of the fourth paragraph) clarifies our argument.

5 P1 … of a scalar field which is found to be dual to the known result .. what do the authors mean by dual. The sentence is too concise to convey any meaning so it should either be expanded on or discussed later. Later on we see that the authors rederive the result of [5] but their result is given in a different form. I would thus say their result is equivalent to that of [5] in this case.

Reply:

Our result is dual, in the sense that the known expression [5] is expressed in terms of bispherical multipoles. However, our approach is based on a round-trip description. Using 'dual' instead of 'equivalent', we wanted to emphasise that our result is not only equivalent to the known one but also allows for an alternative interpretation in terms of round trips.

We have modified the sentence referring to section 3 in the last paragraph of the introduction accordingly. More details concerning this duality are then given later in the paper.

6 P3 Within the scattering approach to the Casimir effect [2], the free energy can be expressed as a Matsubara sum - this is a bit misleading, in any approach the free energy is expressed this way, and it is more common to say a sum over Matsubara frequencies.

Reply:

While we disagree that the free energy is always expressed as a sum over Matsubara frequencies - one could also integrate over real frequencies - we followed the suggestion of the referee and modified the text to avoid using the term 'Matsubara sum'.

7 P3 It would help the reader to give the definition of a Drude type metal where it is mentioned - I think this should be discussed in a detailed way as it is a crucial point in the paper.

Reply:

We extended the first paragraph of section 2. Specifically, we now introduce the dielectric function for a Drude metal and point out that the resulting finite dc conductivity leads to a simplification concerning the modes contributing to the Casimir free energy.

8 P5 Making use of the symmetries of cosine and hyperbolic cosine - I am not sure what is meant by this

Reply:

We were referring to the fact that adding π to the argument of the cosine will merely change the sign and that the hyperbolic cosine is an even function. These properties are of course well known and therefore we refrain from such a detailed discussion in the paper. Since our original hint at the symmetries apparently can constitute a source of confusion, we removed this sentence.

9 P5 again the same mysterious sentence - Casimir free energy which is found to be dual to the known result [5].

Reply:

While we added an explanation of the meaning of duality in the introduction, we have replaced ‘dual’ by ‘equivalent’ in this case.

10 P5 After having determined the matrices associated with the bilinear forms in the exponentials, our main task will be to evaluate the corresponding determinants - I guess the authors mean the integrals in equation 8 which can be written as a determinant ?

Reply:

We do not think that it is necessary to amend the discussion following (9) (previously eq. 8). We trust that the reader knows how to evaluate Gaussian integrals. However, we want to emphasize that the non-trivial step in doing these integrals is to evaluate the determinant of the bilinear form in the exponent.

11 P7 because the sum is converging considerably faster - should be the sum converges faster (I am not sure considerably should be used without more justification).

Reply:

We agree with the statement in parentheses. Since a detailed discussion of convergence properties is beside the main point of the paper, we removed the term ‘considerably’.

12 P7 after eq 19 it would be helpful to explain the difference in boundary conditions for the non expert reader if it has not already be done in response to the point above.

Reply:

In our opinion, the additional explanations at the beginning of section 2 added in response to the referee's point 7 are sufficient to understand the boundary conditions.

13 P7 To obtain the result for the electromagnetic case from the scalar case, we need to determine the contribution of all terms which contain at least one factor −1 when the product in (8) is expanded - I see what this means but it should express more precisely.

Reply:

We have extended the discussion at the beginning of section 4.1 accordingly. We now recall the meaning of the -1 term in (9), previously eq. 8, to clarify how each term accounting for at least one factor -1 in (9), contributes to the monopole part for Dirichlet spheres.

---

## Round 2 · List of Changes

Warnings issued while processing user-supplied markup:

  • Inconsistency: plain/Markdown and reStructuredText syntaxes are mixed. Markdown will be used.
    Add "#coerce:reST" or "#coerce:plain" as the first line of your text to force reStructuredText or no markup.
    You may also contact the helpdesk if the formatting is incorrect and you are unable to edit your text.

List of Changes

1) We have rewritten the first paragraph of the introduction according to requests 2 and 3 of Referee 2. A detailed description of our changes can be found in our reply to the second report.

2) The former third paragraph of the introduction is now split into two paragraphs (three and four in version 2). As a response to the weakness mentioned by the second referee, we extended the discussion of the importance of our result for experiments in paragraph 3. In paragraph 4, we rephrased the role of the zero-frequency term in semi-analytical approaches. For more details, we refer to our reply on the fourth request of referee 2.

3) We have added a paragraph (paragraph 5 in version 2) as a response to request 1 of referee 1, where we discuss in more detail how by combining various results from the literature, our result could be obtained.

4) In paragraph 6, the former fourth paragraph, we extended the motivation for our calculation.

5) In the last paragraph of the introduction, we specified what we mean by 'dual result'. More details can be found in our response to request 5 by referee 2. Furthermore, we added the adjective 'spherical' to the monopole contributions, to avoid confusion with the bispherical multipoles which, in the new version, are now mentioned beforehand.

6) In the first paragraph of section 2, we introduced the definition of a Drude metal by specifying the corresponding dielectric function (see response to request 7 of referee 2).

7) In paragraph two of section 2, we added a remark that the scattering approach to the Casimir effect results in a sum, where a round-trip operator is evaluated at the Matsubara frequencies.

8) Before the old eq. (4), now (5) we removed 'the Wick rotated frequency $\xi$', since we already introduced imaginary frequencies as a consequence of point 6.

9) Before the old eq. (8), now (9) we removed 'Making use of the symmetries of cosine and hyperbolic cosine, we obtain' as a response to request 8 of referee 2.

10) At the beginning of the last paragraph in section 2, we used the singular form for the monopole terms, more precisely, we replaced 'monopole terms do' by 'the monopole term does' and 'monopole terms' by 'monopole term'. This change was made to be consistent with the discussion after eq. (7), now (8). Moreover, we consistently replaced 'monopole contribution' with the plural form 'monopole contributions'.

11) At the end of the last paragraph in section 2, we replaced 'found to be dual' by 'equivalent' (cf. request 9 of referee 2).

12) In the last paragraph of section 3, we specified the possible numerical advantage of our result by replacing 'is' with 'may be numerically'. Moreover, as a response to request 11 of referee 2, we replaced 'the sum is converging considerably faster' by 'it possesses better convergence properties'.

13) We extended the first paragraph of section 4.1, as discussed in our reply to request 13 of referee 2.

14) Before the old eq. (24), now (25) we replaced 'monopole contribution' with 'monopole contributions' (cf. point 10).

15) In the caption of fig. 2, we included a reference to equation (26), former eq. (25). Moreover, we specified the shown block matrix by defining the corresponding values of r and k.

16) Before eq. (27), now (28) and after eq. (32), now (33), we replaced 'monopole contribution' with 'monopole contributions' (cf. point 10).

17) In response to request 2 of referee 1, we extended the discussion in the first part of section 4.4. More precisely, we added the result by Fosco et al., which is now given in eq. (46). Furthermore, we adopted the choice of units by Fosco, a fact now also stated in a footnote, and changed the definitions of the capacitance coefficients (45)-(47), now (47)-(49). Below these definitions, we highlighted the relation between the capacitance coefficients and the round-trip description. We also added the historic reference [27].

18) For consistency with the changes mentioned in point 17, we replaced '$T^2$' by '$R_1R_2T^2$, in the last paragraph of section 4.4.

19) Before eq. (53), now (55), we corrected a typo by replacing 'reads' with 'read'.

20) Before eq. (58), now (60), after eq. (64), now (66) and after eq. (74), now (76), we replaced 'monopole contribution' with 'monopole contributions' (cf. point 10).

21) After eq. (88), now (90) and after eq. (100), now (102) we replaced 'monopole contribution' with 'monopole contributions' (cf. point 10).

---

## Editorial Decision

published